## Perspective

mental health; poverty; social determinants of health; children and adolescents; cash transfer programmes

**Author for correspondence:**
Sara Evans-Lacko,
Email: S.Evans-Lacko@lse.ac.uk

# Potential mechanisms by which cash transfer programmes could improve the mental health and life chances of young people: A conceptual framework and lines of enquiry for research and policy

Sara Evans-Lacko[1] , Ricardo Araya[2], Annette Bauer[1], Emily Garman[3],
Alejandra Álvarez-Iglesias[1,4], David McDaid[1] , Philipp Hessel[5,6],
Alicia Matijasevich[7], Cristiane Silvestre Paula[8], A-La Park[1] and Crick Lund[2,3]

[1]Department of Health Policy, Care Policy and Evaluation Centre, London School of Economics and Political Science, London, UK; [2]Centre for Global Mental Health, Health Service & Population Research Department, Institute of Psychiatry, Psychology & Neuroscience, King's College London, London, UK; [3]Alan J Flisher Centre for Public Mental Health, Department of Psychiatry and Mental Health, University of Cape Town, Cape Town, South Africa; [4]Departamento de Psicología Biológica y de la Salud, Facultad de Psicología, Universidad Autónoma de Madrid, Madrid, Spain; [5]Escuela de Gobierno Alberto Lleras Camargo, Universidad de Los Andes, Bogotá, Colombia; [6]Department of Epidemiology and Public Health, Swiss Tropical and Public Health Institute, Basel, Switzerland; [7]Departamento de Medicina Preventiva, Faculdade de Medicina FMUSP, Universidade de São Paulo, São Paulo, Brasil and [8]Centro Mackenzie de Pesquisa sobre Infância e Adolescência, Programa de Pós-graduação em Distúrbios do Desenvolvimento, Universidade Presbiteriana Mackenzie, São Paulo, Brazil

## Abstract

Mental health is inextricably linked to both poverty and future life chances such as education, skills, labour market attachment and social function. Poverty can lead to poorer mental health, which reduces opportunities and increases the risk of lifetime poverty. Cash transfer programmes are one of the most common strategies to reduce poverty and now reach substantial proportions of populations living in low- and middle-income countries. Because of their rapid expansion in response to the COVID-19 pandemic, they have recently gained even more importance. Recently, there have been suggestions that these cash transfers might improve youth mental health, disrupting the cycle of disadvantage at a critical period of life. Here, we present a conceptual framework describing potential mechanisms by which cash transfer programmes could improve the mental health and life chances of young people. Furthermore, we explore how theories from behavioural economics and cognitive psychology could be used to more specifically target these mechanisms and optimise the impact of cash transfers on youth mental health and life chances. Based on this, we identify several lines of enquiry and action for future research and policy.

## Impact statement

To improve the future life chances of young people from economically deprived backgrounds, we need policies and interventions which target key mechanisms underlying the pathways between poverty and future life chances – in particular, ones which consider youth mental health, which plays a key role in this relationship. Mental health is inextricably linked to both poverty and future life chances, in a vicious cycle. Poverty can lead to poorer mental health, which reduces opportunities and increases the risk of lifetime poverty. Cash transfer programmes are one of the most common strategies to reduce poverty reaching substantial proportions of populations living in low- and middle-income countries. Because of their rapid expansion in response to the COVID-19 pandemic, they have recently gained even more importance. More recently, there have been suggestions that these cash transfers might improve youth mental health, disrupting the cycle of disadvantage at a critical period of life. Here, we present a conceptual framework describing potential mechanisms by which cash transfer programmes could improve the mental health and life chances of young people. Furthermore, we explore how theories from behavioural economics and cognitive psychology could be used to more specifically target these mechanisms and optimise the impact of cash transfers on youth mental health and life chances. Based on this, we identify several lines of enquiry and action for future research and policy.

## Introduction

The majority (61%) of the world's poor are under 24 years of age (Robles Aguilar and Sumner, 2020). Young people living in poverty are disadvantaged across a range of both short- and long-term outcomes (National Academies of Sciences, Engineering, and Medicine, 2019), including decreased future life chances in relation to health, education, relationships and employment, and increased risk of criminal activity. With an additional 97 million people living in poverty in 2020 as a consequence of the COVID-19 pandemic (Gerszon Mahler et al., 2021), a new youth cohort is at increased risk of damaged future life chances.

Mental health is an influential factor intertwined with both poverty (Lund et al., 2018) and future life chances such as education, skills, labour market attachment and social function (Richards et al., 2009). Mental health is defined not only as an absence of mental disorders, but as an asset or a resource that enables positive states of well-being and provides the capability to achieve one's full potential (Patel et al., 2018). Improving young people's mental health can facilitate future life chances and decrease the risk of continuing to live in poverty (Richards et al., 2009; Killackey et al., 2020). Conversely, depression in young people increases likelihood of school failure and reduces chances of future employment and earnings (Clayborne et al., 2019), while youth with conduct problems can cost 10 times more than those without conduct problems because of higher violent offending, drug use, teenage pregnancy and school dropout (Scott, 2015). At the same time, poverty also leads to poor mental health. Young people from economically deprived backgrounds face multiple forms of cumulative disadvantage which significantly limit their life chances and put them at higher risk of mental disorders (Duncan et al., 1998; Ludwig and Miller, 2007; Lund et al., 2011).

Since the 1990s, cash transfer programmes have been widely adopted as a strategy to reduce poverty in low- and middle-income countries (LMICs) (The World Bank, 2018; International Labour Organisation, 2021). Since then, growing research has explored whether these cash transfers might improve youth mental health, disrupting the cycle of disadvantage at a critical period of life (Paxson and Schady, 2010; Lund et al., 2011; Ridley et al., 2020). Some evidence suggests there may be a positive, albeit modest, relationship between cash transfer receipt and improved mental health (see Table 1). Nonetheless, giving money alone does not necessarily lead to better mental health or improved life chances (McGuire et al., 2020; Ziebold et al., 2021; Garman et al., 2022) and some data suggest there can be unintended effects (Fisher et al., 2017) leading to changes in social relations (MacAuslan and Riemenschneider, 2011) and feelings of unfairness which could worsen mental health (Pavanello et al., 2016).

Some of the heterogeneity in findings may be related to the diverse mechanisms by which cash transfers work. Understanding how cash transfer programme characteristics relate to youth mental health, either directly, or indirectly through, for example, changes in parental or youth behaviour, could help us understand how we might better optimise the design and implementation of cash transfer programmes to improve mental health and thus youth life chances – with the potential for wider social and economic benefits (McDaid and Evans-Lacko, 2021).

Here, we present a conceptual framework and discuss potential mediating factors by which cash transfer programmes could improve the mental health and life chances of young people.

Poverty and mental health are intertwined in a complex bidirectional relationship and indeed, some authors have argued that there is a bi-directional causal relationship between poverty and mental health (Ridley et al., 2020). Although the field is still in its infancy, there is growing evidence that at least some thresholds for causality have been crossed. Here, we focus on one piece of the puzzle: the potential for anti-poverty interventions to improve youth mental health. Furthermore, we explore how theories from behavioural economics and cognitive psychology could be used to more specifically target these mediators and optimise the impact of cash transfers on youth mental health and life chances. Our conceptual framework builds on seven existing reviews which examine the impact of cash transfers on mental health (see Table 1). We use the framework described by Ridley et al. (2020) as a starting point. Their paper provides a useful discussion of the individual-level pathways by which anti-poverty programmes affect mental health and some discussion about how contextual factors also have an impact and may therefore moderate the influence of cash transfers on mental health. For example, they highlight worries and uncertainty, environmental factors, physical health, early-life conditions, trauma, violence and crime, and social status, shame and isolation. Building on this, we take into account other recent related reviews and frameworks which discuss the social determinants of mental health, and which highlight additional societal-level mechanisms which may mediate or moderate the relationship between cash transfers, youth mental health and future life chances. For example, we know macro-level factors such as social capital, social cohesion, income inequality and macro-economic shocks can influence mental health (Stuckler et al., 2009; Ehsan and de Silva, 2015; Ribeiro et al., 2017; Lund et al., 2018; Patel et al., 2018) and research also suggests that cash transfer programmes also influence these societal level outcomes (Veras Soares et al., 2006; Bastagli, 2010; Loureiro, 2012; McKnight, 2015; Drucza, 2016; Machado et al., 2018; Breckin, 2019). Based on this, we identify several lines of enquiry and action for future research and policy.

## How might cash transfers influence youth mental health and life chances?

Figure 1 depicts the hypothesised relationships between cash transfer programmes, youth mental health and life chances outcomes. It identifies potential mediators which could optimise the impact of cash transfer programmes for youth mental health and in turn lead to improved life chances. For this paper, we conceptualise life chances as social and economic opportunities available to someone depending on their circumstances or context (Roth, 1981; Evans et al., 2000) and which are shaped by structural factors. Based on previous work of Richards et al. (2009) on youth mental health and life chances, we focus on life chances related to education, skills and labour market attachment and social function. The framework is relevant for young people aged 10 to 24 years. We select this age group and exclude very young children so that we can focus more on targeted mechanisms present in young people and adolescents. This is in line with more recent definitions of adolescence and young people which extend to a slightly older age as this reflects better current understanding of patterns of adolescent development (Patton et al., 2016). Furthermore, this also represents the age group for which mental disorders represent the primary cause of disability worldwide (Armocida et al., 2022).

**Table 1.** Summary of key systematic reviews and meta-analyses linking cash transfer impacts on mental health

| Reference (by date published) | Aims | No. of studies | No. of studies focused on youth | Mental health outcomes considered in included papers | Key findings | Effect sizes and overall effects | Proposed mechanisms by which cash transfer impacts mental health |
|---|---|---|---|---|---|---|---|
| Lund et al. (2011) | Two systematic reviews: Review 1: To assess the effect of poverty alleviation interventions on mental, neurological and substance misuse disorder outcomes in LMICs Review 2: To assess the effect of mental health interventions on individual and family or carer economic status in these countries | Review 1: 5 Review 2: 9 | Review 1: 4 Review 2: 0 | Depressive symptoms Cognitive development Behavioural problems Self-esteem* | The mental health effect of poverty alleviation interventions was inconclusive, although some CCT and asset promotion programmes showed mental health benefits Some indications that CCTs and asset promotion are more clearly associated with mental health benefits than other poverty alleviation interventions | For both reviews, heterogeneity precluded an attempt to draw summary estimates of effect size Review 1: 3/5 (60%) studies did not show significant treatment effects Review 2: 10/19 (53%) associations tested showed the intervention to have a significant positive effect on economic status. No study showed a mental health intervention to have a significant negative effect on economic status | 'With respect to causal mechanisms, the scarce evidence for poverty alleviation interventions with a financial component do not allow strong conclusions… Evaluations that include an analysis of separate components of the interventions might contribute to a clearer picture – e.g., whether the regularity of payments or inputs, their conditionality, or their cumulative amounts are key factors determining mental health outcomes' |
| Ridley et al. (2020) | To review the interdisciplinary evidence of the bi-directional causal relationship between poverty and common mental illnesses To review the impacts of anti-poverty programmes on mental health | 30 | 14 | 'Common mental illnesses' (depressive and anxiety disorders) | Evidence from natural experiments confirms the causal relationship between mental illness and poverty Among the mechanisms, worries and uncertainty and the environment (e.g., pollution) are key for poverty causing mental illness; and conversely beliefs, preferences and cognitive function serve as mental illness mechanisms that cause poverty | The overall impact of anti-poverty programmes was 0.094 SD (95% CI: 0.040, 0.147) The overall impact of anti-poverty programmes when all available follow-up measures were included was 0.109 SD (95% CI: 0.065, 0.153) | It is proposed that anti-poverty programmes, including – CTs, could reduce uncertainty associated with volatile income and expenditure and smooth economic shocks. Effectively reducing poverty may also mitigate exposure to environmental stressors such as violence and associated trauma, improve parent mental health and potentially improve early-life condition, including access to nutrition, resulting in improved cognitive development all important risk factors for future mental illness |
| McGuire et al. (2020) | To evaluate whether CTs improve SWB and mental health among recipients in LMICs | 45 | Not specified | SWB measures: life satisfaction, happiness Mental health measures: CESD-10, CESD-20, GSD-15, worry and anxiety, GHQ-12, distress, depression, SF-12, SRQ-20, MHI-5, K-10 | After an average follow-up time of 2 years, CTs had a small but statistically significant positive effect on both SWB (Cohen's $d = 0.13$, 95% CI: 0.09, 0.18) and mental health ($d = 0.07$, 95% CI: 0.05, 0.09) among recipients | CTs had a small but statistically significant positive effect on both SWB (Cohen's $d = 0.13$, 95% CI: 0.09, 0.18) and MH ($d = 0.07$, 95% CI: 0.05, 0.09) among recipients CT value, both relative to previous income and in absolute terms, was a strong predictor of the effect size | Explaining specific mechanisms by which CTs impact mental health and well-being was beyond the scope of the review; however, it noted potential pathways may include: improvement in food security and reduction in stress associated with financial instability and hardship in short run. In the long-term, CTs could provide a feeling of economic security to foster improved social relationships to allow access to education and focus on future investments Further by protecting against future economic shocks, recipients could focus on long-term goals rather than immediate survival needs |
| Zimmerman et al. (2021) | To review the literature on the effect of cash transfers | 12 | 12 | Included but not limited to the following disorders: | While cash transfers may positively impact mental | 11/13 interventions (85%) showed a significant | Although not directly evaluated, it proposed that CTs could directly reduce financial |

**Table 1.** (*Continued*)

| Reference (by date published) | Aims | No. of studies | No. of studies focused on youth | Mental health outcomes considered in included papers | Key findings | Effect sizes and overall effects | Proposed mechanisms by which cash transfer impacts mental health |
|---|---|---|---|---|---|---|---|
| | programmes on the mental health of children and young people (0–24 years old) in LMICs. To understand whether different types of cash transfer programmes have different effects on children and young people's mental health | | | mood, anxiety, PTSD, substance-related, feeding and eating, psychotic, personality, behavioural problems, self-esteem, confidence, resilience, self-efficacy, future outlook and hopefulness | health, the heterogeneity across studies and mental health outcomes suggests that their effects are likely to depend on the social, cultural and economic context in which they are implemented, as well as on their design and the role of conditionality<br>There is a need for high-quality RCTs to assess the impact of poverty reduction interventions on mental health in children and young people | positive impact of CT on at least one mental health outcome in children and young people.<br>The meta-analysis showed no impact of cash transfers on depressive symptoms (0.02, 95% CIs: −0.19 to 0.23; $p = 0.85$) | strain and increase economic security through increased income. They may also reduce family conflict associated with poverty and financial stress, thus reducing mental health risks for all family members. CTs may also reduce child labour and related exposures that place young people at risk of mental health conditions |
| Romero et al. (2021) | To synthesise the evidence on the causal impact of economic transfers on mental health and subjective well-being | 57 | 14 | Three groups: depression, stress or anxiety and happiness or life satisfaction | Positive effects of cash transfers (especially unconditional ones) on depression, happiness and life satisfaction<br>However, limitations come from the relatively small sample size and the overall heterogeneity of effect sizes, with an estimate of $I^2 = 93.1\%$ | Economic interventions had a positive effect on well-being: on average, an intervention increased well-being by 0.100 standard deviations (SDs). The largest impacts occurred for asset transfers (0.158 SD) and unconditional cash transfers (0.150 SD)<br>There is no clear relationship between effect size and transfer magnitude, possibly due to heterogeneity in samples | This review did not discuss or assess potential mechanisms of CTs on mental health |
| Zaneva et al. (2022) | To provide causal evidence that monetary interventions reduce internalising symptoms of adolescents experiencing poverty | 14 | 14 | Focus on internalising symptoms (including depression, anxiety, trauma and mental distress)<br>Excludes broader nonclinical outcomes like well-being, happiness, or life satisfaction | While CT programmes are generally effective in improving mental health of adolescents, conditionality for girls in CCTs can add a burden in low- and middle-income countries, and as such may constitute a source of stress and may produce worse mental health outcomes | Meta-analysis (on eight of the studies) revealed that internalising problems were significantly reduced post-intervention compared to control (OR = 0.72, 95% CI: 0.59–0.88, $p < 0.01$; $I^2 = 67\%$, $\tau^2 = 0.05$, $p < 0.01$) | The paper proposed potential mechanisms related to differential gender effects of CTs including economic autonomy and that boys have more autonomy and are more likely to spend transferred money as they like, compared to girls who may feel pressure to contribute to the family budget. It was also noted that giving the money to parents versus youth could lead to different causal mechanisms |

Abbreviations: CCT, conditional cash transfer; CESD, Centre for Epidemiologic Studies Depression Scale; CI, confidence interval; CT, cash transfer; CTP, cash transfer programme; GDS, Geriatric Depression Scale; GHQ, General Health Questionnaire; $I^2$, statistic describing the percentage of variation across studies that is due to heterogeneity rather than chance; K-10, Kessler Psychological Distress Scale; LMICs, low- and middle-income countries; MHI-5, 5-Question Mental Health Inventory; PPP, purchasing power parity; PTSD, post-traumatic stress disorder; RCT, randomised controlled trial; SF-12, 12-Item Short Form Survey; SRQ-20, 20-Item Self Reporting Questionnaire; SWB, subjective wellbeing.*Mental health outcome assessed among youth participants (aged 10–24)

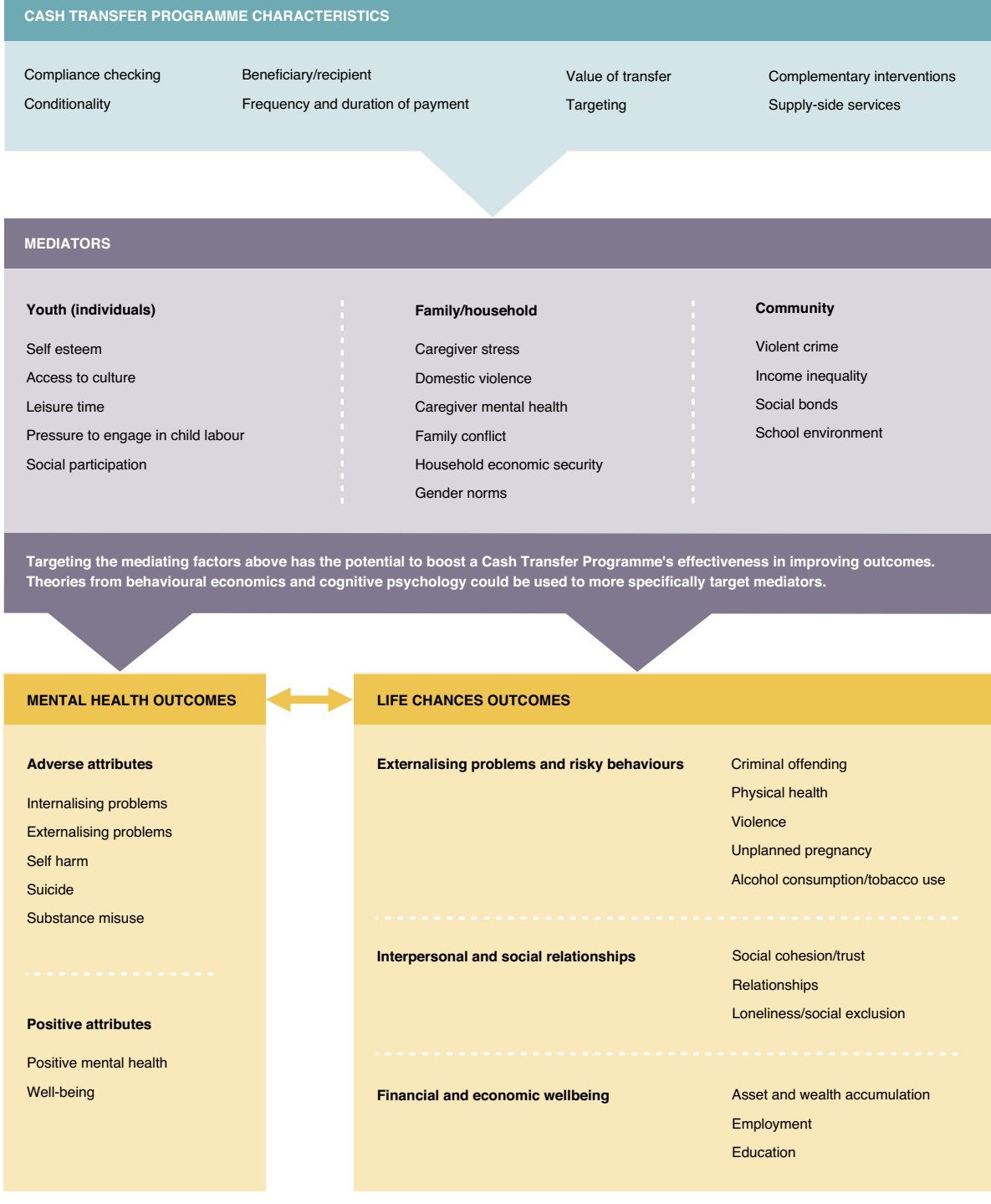

**Figure 1.** The influence of cash transfer programmes on youth mental health and life chances and potential mechanisms for optimising youth outcomes.

## Cash transfer programme characteristics

Cash transfer programmes vary in relation to a number of characteristics including the *transfer value, frequency and duration of money* provided; presence and rigidity of *conditionalities;* *compliance checking* (including benefit restrictions); *targeting* (*who receives the money* in the household and eligibility restrictions); *payment mechanisms for how* the cash is accessed and *supply-side services*. All of these characteristics may affect whether or not, and to what degree, cash transfers affect youth mental health. For

example, a recent meta-analysis suggests that the value of the cash transfer, both in relation to previous income and in absolute terms, is a strong predictor of the size of the effect on recipient's mental health (McGuire et al., 2022).

### Youth mediators

There are a number of potential youth mediators through which cash transfers could indirectly influence youth mental health. In Malawi, young people reported being ashamed to go out in old, tattered clothing and the increased expenditure enabled by cash transfers allowed them to buy clothing and thereby improve *self-esteem* (Baird et al., 2013). Reduced shame and increased self-esteem may also lead to increased participation in *social* (MacPhail et al., 2018; Bastagli et al., 2019; Pozuelo et al., 2020) and *cultural activities* and improved confidence and status in peer networks, which hence impact on mental health. Cash transfers are also associated with a reduction in forced *child labour* (Bastagli et al., 2019), particularly with reduced intensity of labour and/or number of hours worked, which may previously have provided an economic cushion for households.

### Family/household mediators

At the family/household level, cash transfers, particularly when given to the female head of the household, can lead to improvements in well-being of its members and reductions in household stress and conflict (Bardasi and Garcia, 2015). Lower *caregiver stress* resulting from cash transfer receipt may decrease the need to call upon negative coping strategies, such as alcohol consumption, and reduce *domestic violence* (Borraz and Munyo, 2020; Ohrnberger et al., 2020), thereby improving youth mental health (Costello et al., 2010). Increased *household economic security* and emotional well-being directly resulting from the infusion of cash can also lead to improved youth mental health (Buller et al., 2018; Eyal and Burns, 2019). Cash transfer programmes can also reduce financial arguments between parents and improve financial coping strategies. Targeting cash transfer programmes at mothers may also increase their self-esteem and perceived value in the household. There is, however, some evidence of unintended consequences for the last two pathways suggesting that *gender norms* and dynamics in the community or household should be taken into account in relation to how the cash transfer is framed, particularly so that the male head of household does not feel that their role or control of finances is threatened (Buller et al., 2018). For example, a randomised controlled trial (RCT) of a cash and in-kind food transfer programme in Ecuador reduced physical and sexual violence by 30%. It was suggested that linking the transfer to child nutrition was important as this was perceived to be the mothers' responsibility and men did not feel challenged (Hidrobo et al., 2016).

### Community mediators

Although cash transfer programmes are targeted at individuals or households, impacts on youth and families can aggregate into macro level effects on communities. Cash transfer programmes provide an income boost to the poorest individuals and can thus reduce poverty and narrow income inequality (Veras Soares et al., 2006; Bastagli, 2010; McKnight, 2015). Cash transfer programmes may also lead to reductions *in community violence and social exclusion* (Drucza, 2016), partly through increased

social trust and strengthening *social bonds.* Reduced poverty and income-inequality (Loureiro, 2012) and stronger social bonds (Breckin, 2019) have all been posed as pathways whereby cash transfers can reduce community violence (Machado et al., 2018). Given the importance of poverty, income inequality and community violence as social determinants of mental health (Lund et al., 2018), these represent other potentially important pathways by which cash transfers could impact youth mental health and life chances.

Many of these potential mediators at the community level also represent *contextual factors*, which could act as effect modifiers. For example, homicide rates and income inequality which are linked with mental health are also reduced by cash transfer programmes (Machado et al., 2018). Thus, the effects of cash transfers may also vary according to the presence of contextual factors including relative poverty and disadvantage, levels of income inequality, violence and unemployment (Owusu-Addo et al., 2018). Moreover, where conditionalities are included, for example, based on school attendance (one of the most common conditionalities), effects on youth mental health may depend on a range of contextual variables including school quality, costs of attending school, presence of bullying and academic performance.

### Mental health and consequences for life chances

There is a good deal of evidence that improved youth mental health is associated with more positive life chances outcomes, including for physical health (Thompson et al., 2021), education (Dalsgaard et al., 2020; Hoffmann et al., 2021) and employment (Thompson et al., 2021), in the short-term and through to mid-life. The potential for cash transfers to influence mental health and life chances could thus facilitate a virtuous cycle. Youth emotional and behavioural problems independently predict outcomes in adult life, such as social class and social adjustment (Caspi et al., 2020; Laceulle et al., 2020; Thompson et al., 2021). In particular, youth *externalising problems* such as conduct problems are more strongly associated with poor life chances over the lifetime in comparison with internalising problems (Richards et al., 2009; Knapp and Evans-Lacko, 2015; Hammerton et al., 2019). Finally, analysis across the 1946, 1958 and 1970 British birth cohorts (Colman et al., 2009; Richards et al., 2009; Caspi et al., 2020; Gronholm et al., 2022) found that childhood conduct problems were associated with lower educational qualifications, persistent economic inactivity, lower earnings and increased criminal convictions and arrests.

### Could more attention to mechanisms in the design and implementation of cash transfer programmes improve youth mental health outcomes?

Although evidence suggests there is potential for cash transfer programmes to improve youth mental health (Zimmerman et al., 2021), cash transfers are not a panacea and on their own tend to be a relatively 'blunt' instrument, which, while providing additional financial resources to the household, were never designed to improve youth mental health. Because cash transfer programmes do not necessarily address the mechanisms by which poverty undermines mental health, we may not fully realise the potential of these programmes to impact youth mental health. More attention to the cognitive, affective, behavioural and contextual barriers at the youth, family and community levels in the design and

implementation of cash transfers could potentiate their benefits for youth mental health and life chances.

Theories from behavioural economics and cognitive psychology can offer some useful insights about behaviour change which could inform the operation of cash transfer programme policies to improve youth mental health and life chances (Gennetian et al., 2021). Behavioural economics would suggest that we use positive reinforcement to incentivise desired behaviours rather than punishments or mandates (or rigid conditionalities) which penalise individuals who do not comply with specified conditions or policies and may undermine youth mental health (Thaler and Sunstein, 2008). Nudge theory, fresh start effects and thinking slow, for example, have been explored in other studies and may offer potentially useful ideas about how cash transfer programmes could facilitate youth mental health and life chances. We summarise the considerations for potential design and implementation strategies here.

*Nudge theory* posits that the choice architecture shaping one's environment and the framing of decisions can positively influence decision-making, allowing individuals to make better decisions in line with their goals (Thaler and Sunstein, 2008). A key facet of this theory is that individuals need to maintain some feeling of control or autonomy in their decision and that the desired behaviour is positively incentivised rather than mandated or enforced. Some of the cognitive and affective consequences of poverty may make it difficult for people to be intentional with their spending and align it with future goals and aspirations and hence limit life chances. Poverty, for instance, is hypothesised to reduce executive functioning, thus increasing future discounting, impairing decision-making, and inducing negative affective states such as anxiety and depression (Haushofer and Fehr, 2014). Given the cognitive burden that individuals living in poverty already face, complex administrative procedures and rigid conditionalities which penalise individuals for not meeting required conditions may also lead to increased household stress and reduce feelings of self-efficacy, subsequently negatively impacting on caregiver and youth mental health and life chances (Baird et al., 2013). During the COVID-19 pandemic, many programmes have become more flexible with administrative procedures and also relaxed conditionalities providing a template for testing alternatives (Bauer et al., 2021).

Spending decisions could also be aided by framing messages about what the cash is for and what could be done with it. For example, a labelled cash transfer programme in Morocco which was framed as an education support programme, improved parents' beliefs about the value of education and greatly increased school participation (Benhassine et al., 2015). Messaging around how the cash could be used to target mediating factors important for youth mental health such as social participation or access to culture could lead to greater impacts on youth mental health.

*Fresh start effects and temporal landmarks* can also be used to motivate aspirational behaviour (Dai et al., 2014). In line with this theory, days or events can be identified or framed as a fresh start moment – a period when individuals tend to be more open and interested in striving to achieve goals or ambitions. Research suggests that identifying set days such as birthdays or a new year could act as a fresh start; cash transfer programmes could recognise the timing of the cash transfer delivery as a window by which individuals may be more open to long-term planning and thus potentially more successful at overcoming the various barriers that they face. One cash transfer programme from Madagascar, for example, supports mothers who are the recipients of the cash transfer to consider longer-term planning around expenses and provides positive incentives around investment just before they receive the monthly payments (Vermehren et al., 2019). This type of timed nudging may break the cycle whereby individuals tend to consider immediate expenses and could also engender a feeling of self-efficacy and self-control.

Finding ways to incorporate *thinking slow* and deliberately rather than *thinking fast* and automatically (Kahneman, 2011; Heller et al., 2017), could reduce impulsivity and allow individuals to be more considered in their reactions to various situations and events, particularly challenging ones which could lead to conflict and harm youth mental health and life chances. We know that youth living in poverty face several environmental and contextual challenges including high levels of violence, family instability, high levels of unemployment and poor quality schools. These factors are associated with increased mental health problems and likely reduce the potential for cash transfer programmes to improve mental health and life chances. In particular, 'cash plus' programmes might offer cognitive behavioural therapy (CBT) or other types of psychological support which could help youth and parents of youth to learn strategies for thinking more slowly and deliberately. For example, combining cash transfers with CBT reduced disruptive behaviour problems among young men involved in criminal activities in Liberia (Blattman et al., 2017) more effectively than either intervention on its own.

While integrating theories from economics and cognitive psychology represents a promising approach, more research is needed to better understand cash transfer design and implementation factors in relation to youth mental health and how the contextual characteristics of population and setting may mediate and/or moderate outcomes (Skivington et al., 2021). For example, a recent trial in Kenya did not find an interaction between psychotherapy (in this case, problem management plus [PM+]) and cash transfer receipt. Moreover, at 12 months follow-up, when looking at each intervention on its own, only those receiving the cash transfer had positive impacts on mental health. Given the large amount of evidence demonstrating positive impacts of psychosocial interventions in low resource settings (Barbui et al., 2020), this suggests that we need a better understanding of how these complex combined interventions work together and how they interact with those who are receiving the intervention, the context in which it is delivered and what interventions might be needed to potentiate positive impacts.

## Future directions for research and policy

The impacts of cash transfer programmes can go beyond those of their specific objectives. Providing cash transfers to a household as a means of reducing poverty could also improve youth mental health through a range of individual youth, family/caregiver and community mediators. Nonetheless, cash transfer programmes alone are likely to be insufficient to improve youth mental health and life chances (Dai et al., 2014; Zimmerman et al., 2021). To more effectively improve youth mental health and life chances outcomes, more specific targeting of youth mental health and associated mechanisms need to be incorporated into programme design and implementation. This requires better understanding of the potential mechanisms along the pathway to improved youth mental health and life chances and consideration of behavioural economics and cognitive theories.

To address these questions, future research could consider three avenues of enquiry: (I) Greater attention could be given to conducting mediation and moderation analysis to examine the

mechanisms by which cash transfer programmes improve youth mental health, in particular considering how these interventions may interact with individual youth, household and community contextual characteristics. Thus, evaluations should also acknowledge the complexity in the relationship between poverty and mental health, which may vary substantially depending on the mental health condition and the specific contextual characteristics of poverty (Dandona et al., 2018; Mor et al., 2018; Juárez et al., 2019). Some contextual factors such as urbanicity may act as both risk and protective factors for mental health conditions (Solmi et al., 2017) and the nuances of these potential moderators could help us better understand the role of cash transfers to improve youth mental health. This requires using large longitudinal cohorts which assess youth and parent factors, linked with other contextual community data. (II) Consideration could be given to a wider range of mental health outcomes which may be impacted by cash transfers. Existing evaluations which have considered mental health tend to focus on depressive symptoms or psychological distress with less investigation of impacts on externalising problems such as conduct disorder. Moreover, in addition to pathological aspects of mental health problems, mental health should be considered as a continuum, including measurement of functioning, self-efficacy and well-being (Westerhof and Keyes, 2010; Patel et al., 2018) which can be important for capabilities (Sen, 2009). (III) Strategic multidisciplinary collaborations are needed to facilitate understanding of the complex pathways by which cash transfers interact to influence youth mental health and life chances. Bringing together researchers from economics, social science, anthropology, psychology and neuroscience can facilitate data collection, analysis and interpretation of the social, economic and neurobiological determinants of mental health and associated life chances.

In relation to policy: (I) Further work should be done to design cash transfer programmes that target mechanisms which specifically support youth mental health and well-being or an evaluation which also considers broader social and mental health impacts on youth (MacAuslan and Riemenschneider, 2011; Fisher et al., 2017). (II) To be most effective, cash transfer programmes should consider the underlying sociocultural, economic and political context and framing conditionalities which are sensitive to that context; for example, by relaxing restrictions and conditionalities during the COVID-19 pandemic (Bauer et al., 2021). (III) 'Cash transfer plus' programmes should be considered by policy makers, incorporating evidence-based youth mental health treatment or prevention interventions into cash transfer programmes. These types of programme developments may require piloting to reduce any unintended effects (e.g., so as not to highlight the connection between poverty and mental health problems in a way that could increase prejudice or internalised stigma of beneficiaries in a cash 'plus' programme). Moreover, additional costs and benefits should be weighed carefully given the reality of constrained resources. For example, we acknowledge that there is a risk to investing in cash transfer 'plus' programmes without essential complementary services such as free basic healthcare, education and social services, especially in LMICs where these are typically under-resourced and this should be further considered when piloting and evaluating new strategies.

Cash transfer programmes are likely to continue to be used in many countries for the foreseeable future, regardless of how effective they are for mental health. Thus, it is important to try to understand how to potentiate their benefits for youth mental health alongside other actions to improve mental health. In this vein, it is important to consider how we might incorporate youth mental health promotion, prevention (World Health Organization, 2020) and treatment interventions to complement cash transfer programmes and more specifically target youth mental health and associated mechanisms (Dai et al., 2014).

**Open peer review.** To view the open peer review materials for this article, please visit http://doi.org/10.1017/gmh.2023.4.

**Acknowledgements.** CHANCES-6 is a study of the effect of cash transfers on youth mental health and life chances in Brazil, Colombia, Liberia, Malawi, Mexico and South Africa (2018–2022). This paper reflects on empirical findings from analysis of youth cohorts in the six countries, qualitative research in Brazil, Colombia and South Africa; and engagement with local and international stakeholders in youth mental health and poverty.

**Author contributions.** S.E.-L. and C.L. conceptualised the manuscript. R.A.B., A.B., E.G., A.A.-I., D.M., P.H., A.M., C.S.P. and A.P. contributed to the drafting and editing of the manuscript.

**Financial support.** This work was supported by the UKRI's Global Challenges Research Fund (ES/S001050/1). The support of the Economic and Social Research Council (ESRC) is gratefully acknowledged.

**Competing interest.** We declare no competing interests.

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
