## [Reviewer Report]

Gary Belkin

Cambridge Prisms: Global Mental Health

12 December 2022

Dear Professor Belkin and Cambridge Prisms: Global Mental Health Editorial Board,

It brings me great pleasure to submit to you for review our manuscript entitled “Potential Mechanisms by Which Cash Transfer Programmes Could Improve the Mental Health and Life Chances of Young People: A Conceptual Framework and Lines of Enquiry for Research and Policy” for publication as a Perspective in Cambridge Prisms: Global Mental Health. 

To improve future life chances of young people from economically deprived backgrounds, we need policies and interventions which target key mechanisms underlying the pathways between poverty and future life chances—in particular ones which consider youth mental health which plays a key role in this relationship. Mental health is inextricably linked to both poverty and future life chances, or factors which reduce future quality of life, in a vicious cycle. Poverty can lead to poorer mental health, which reduces opportunities and increases the risk of lifetime poverty. 

Cash transfer programmes are one of the most common strategies to reduce poverty reaching substantial proportions of populations living in low- and middle-income countries. Because of their rapid expansion in response to the Covid-19 pandemic, they have recently gained even more importance.

More recently, there have been suggestions that these cash transfers might improve youth mental health, disrupting the cycle of disadvantage at a critical period of life. 

Here, we present a conceptual framework describing potential mechanisms by which cash transfer programmes could improve the mental health and life chances of young people. Further we explore how theories from behavioural economics and cognitive psychology could be used to more specifically target these mechanisms and optimise the impact of cash transfers on youth mental health and life chances. Based on this, we identify several lines of enquiry and action for future research and policy. 

We hope you find this manuscript intriguing and worthy of publication in Cambridge Prisms: Global Mental Health. If you have any questions or concerns, please feel free to contact the corresponding author: Sara Evans-Lacko.

Thank you for your time and I look forward to hearing from you.

Sincerely,

Sara Evans-Lacko, PhD

On behalf of co-authors, Ricardo Araya, Annette Bauer, Emily Garman, Alejandra Álvarez-Iglesias, David McDaid, Philipp Hessel, Alicia Matijasevich, Cristiane Silvestre Paula, A-La Park, Crick Lund and the CHANCES-6 study team

Corresponding author contact information:

Sara Evans-Lacko, Care Policy and Evaluation Centre, London School of Economics and Political Science, Houghton Street, London, UK, WC2A 2AE.

Phone: (0)20 7405 7686

Email: S.Evans-Lacko@lse.ac.uk

---

## [Reviewer Report]

*Comments to Author*: Studying potential mechanisms by which cash transfer programs could improve young people’s mental health is an important issue. In part, one wants to be sure that there is no unintended harm but also because, unlike in a developed country context, cash transfers compete for resources with the funds needed for even the minimal provision of essential services such as public health, healthcare, and education. There is also the concern that since in many of these essential service areas, markets do not work well, cash transfers could lead to increased consumption of, say, unnecessary antibiotics or hospitalisation and jukus (cram schools) instead of the more desirable investments in preventive healthcare or higher quality schools. Cash transfers are politically attractive because of their clientelist nature and because they absolve public officials from the need to build public services. Therefore, as the paper seeks to do, it is essential to provide robust guidance to policymakers on the circumstances/prerequisites that need to be in place for them to have an impact.

The authors review several studies to build their framework, which links program characteristics with mediators and moderators of impact on youth mental health. This is a very useful framework and draws attention to the importance of the complementary actions that need to accompany cash transfer programs if they are to result in the maximum possible impact. In the current version of the paper, the authors, however, don’t dwell on how, in the absence of even minimal service capabilities in most developing environments, these complementary services may be provided, nor do they clearly state what the implications for cash-transfer programs would be if these complementary services are missing. Perhaps they may wish to examine this in greater depth in future versions.

I also recommend that the authors phrase their statements of links between ill-defined concepts such as poverty and mental health much more carefully. Their current language may imply a strong causal linkage, whereas my understanding of the evidence is that most findings are associational and articulated as increases in risk. Causality, where it does exist, is linked to multiple thresholds that need to be crossed. There is a real danger in my mind otherwise of further stigmatisation of the poor, particularly in environments like India, where while the proportion of people below $2 a day may have fallen considerably, the ratio below the $10 (middle class) cut-off may be as high as 97% (Mor et al., 2018).

It is also important to bear in mind that some aspects associated with poverty, such as overcrowding, have mixed effects. For example, while for schizophrenia and related psychotic disorders, urban environments appear to have a distinct pathogenic effect (Solmi et al., 2017), for depressive disorders and suicides, the effect appears to be protective (He et al., 2020; Solmi et al., 2017). Another example is that while in the US, poverty rates amongst Black Americans, at 31.7%, are much higher than the 18.2% for White Americans, on account of many factors, the level of optimism amongst Black Americans is much higher (Graham & Pinto, 2018). The higher level of optimism perhaps results in lower feelings of “defeat and humiliation” (O’Connor & Kirtley, 2018) amongst them and, because of that, significantly lower rates of suicide. There are also societal and familial factors that, for example, account for the fact that suicide mortality rates are low in the poorest Indian states, such as Bihar and much higher in the wealthiest states, such as Kerala and Tamil Nadu (Dandona et al., 2018).

References

Dandona, R., Kumar, G. A., Dhaliwal, R. S., Naghavi, M., Vos, T., Shukla, D. K., Vijayakumar, L., Gururaj, G., Thakur, J. S., Ambekar, A., Sagar, R., Arora, M., Bhardwaj, D., Chakma, J. K., Dutta, E., Furtado, M., Glenn, S., Hawley, C., Johnson, S. C., … Dandona, L. (2018). Gender differentials and state variations in suicide deaths in India: the Global Burden of Disease Study 1990-2016. The Lancet Public Health, 3(10), e478--e489. https://doi.org/10.1016/s2468-2667(18)30138-5

Graham, C., & Pinto, S. (2018). Unequal hopes and lives in the {USA}: optimism, race, place, and premature mortality. Journal of Population Economics, 32(2), 665–733. https://doi.org/10.1007/s00148-018-0687-y

He, S., Song, D., & Jian, W. (2020). The Association between Urbanization and Depression among the Middle-Aged and Elderly: A Longitudinal Study in China. INQUIRY: The Journal of Health Care Organization, Provision, and Financing, 57, 0046958020965470. https://doi.org/10.1177/0046958020965470

Mor, N., Ahluwalia, R., & Atmavilas, Y. (2018). To address inequality, India must address underlying factors of poverty. India Development Review. https://idronline.org/addressing-inequality-in-india/

O’Connor, R. C., & Kirtley, O. J. (2018). The integrated motivational-volitional model of suicidal behaviour. Philosophical Transactions of the Royal Society B: Biological Sciences, 373(1754), 20170268. https://doi.org/10.1098/rstb.2017.0268

Solmi, F., Dykxhoorn, J., & Kirkbride, J. B. (2017). Urban-Rural Differences in Major Mental Health Conditions. In N. Okkels, C. B. Kristiansen, & P. Munk-Jorgensen (Eds.), Mental Health and Illness in the City. Mental Health and Illness Worldwide (pp. 1–106). Springer Singapore. https://doi.org/10.1007/978-981-10-0752-1_7-1

---

## [Reviewer Report]

*Comments to Author*: This is a comprehensive and conceptually-driven review of the expansive literature on cash transfer programs and their impact on youth—the latter defined broadly in terms of age grouping.

The authors adopt a global perspective and emphasize the impacts of cash transfers not only on the affected individuals but on households and communities. The growing number of such programs makes this a daunting task and clearly ‘local’ factors including cultural norms deserve close attention. Perhaps not surprisingly, the effects of cash transfers on varied outcomes including mental health were weakly connect and the authors suggest larger longitudinal studies to disentangle mediating and moderating effects.

This review is straightforward and responsible in identifying positive/negative/neutral impacts

of cash transfers based upon a sweeping ‘capture’ of extant literature. It is possible that new conceptual frameworks simply focus on areas of plausible impact and not harness cash transfers with complex phenomena such as mental health and stigma.

---

## [Reviewer Report]

*Comments to Author*: The article (like the reviewers indicate) is an important piece of work that focusses on responding to the crisis of the oft repeated and yet infrequently addressed social determinants of mental health. The lead up to the deconstruction of factors that influence social health / mental health , followed by interventions that may support impactful outcomes in multiple domains builds on pre existing frameworks - this works well and supports the need for continued work in this direction ( longitudinal studies etc. outlined in the article). While methods and approaches from Behavioural economics and psychology have been highlighted, the importance of socio- political systems and structures may perhaps be emphasised a little more ( if possible in this article, and if not, in future). Change is often not exclusively linked to individuals, but to ecosystems and structures ; therefore challenging structural barriers assume importance like the authors have indicated using a social and economic intervention. I would therefore look for long term granular level positive and progressive social changes or ( the absence of the same ) to better understand in-depth processes that facilitate these -at micro, meso and macro levels using ethnography. Understanding every day changes in decision making that impact self- esteem and self- efficacy may further help deconstruct what we cluster under broader categories of mental health, social determinants and life changes.

---

## [Reviewer Report]

*Comments to Author*: This is a comprehensive and conceptually-driven review of the expansive literature on cash transfer programs and their impact on youth—the latter defined broadly in terms of age grouping.

The authors adopt a global perspective and emphasize the impacts of cash transfers not only on the affected individuals but on households and communities. The growing number of such programs makes this a daunting task and clearly ‘local’ factors including cultural norms deserve close attention. Perhaps not surprisingly, the effects of cash transfers on varied outcomes including mental health were weakly connected and the authors suggest larger longitudinal studies to disentangle mediating and moderating effects.

This review is straightforward and responsible in identifying positive/negative/neutral impacts

of cash transfers based upon a sweeping ‘capture’ of extant literature. It is possible that new conceptual frameworks simply focus on areas of plausible impact and not harness cash transfers with complex phenomena such as mental health and stigma.

---

## [Reviewer Report]

*Comments to Author*: The paper reads very well now. It is useful to review prospects of cash transfers/ cash transfers impacting mental health gains. Towards this end, this paper synthesises relevant and important information , even as it develops newer frameworks, taking into account nuances whose essentiality in this context cannot be emphasised enough.